# Genetics of white color and iridophoroma in "Lemon Frost" leopard geckos

**Longhua Guo**[1]*, **Joshua Bloom**[1], **Steve Sykes**[2], **Elaine Huang**[3], **Zain Kashif**[1], **Elise Pham**[1], **Katarina Ho**[1], **Ana Alcaraz**[4], **Xinshu Grace Xiao**[3], **Sandra Duarte-Vogel**[5], **Leonid Kruglyak**[1]*

**1** Department of Human Genetics, Department of Biological Chemistry, Howard Hughes Medical Institute, University of California, Los Angeles, California, United States of America, **2** Geckos Etc. Herpetoculture, Rocklin, California, United States of America, **3** Department of Integrative Biology and Physiology, University of California, Los Angeles, California, United States of America, **4** College of Veterinary Medicine, Western University of Health Sciences, Pomona, California, United States of America, **5** Division of Laboratory Animal Medicine, David Geffen School of Medicine, University of California, Los Angeles, California, United States of America

* longhuaguo@mednet.ucla.edu (LG); lkruglyak@mednet.ucla.edu (LK)

**Data Availability Statement:** All sequencing data is available from the NCBI SRA database (accession number PRJNA730084).

**Funding:** This work is supported by the Howard Hughes Medical Institute (to LK) and the Helen Hay

## Abstract

The squamates (lizards and snakes) are close relatives of birds and mammals, with more than 10,000 described species that display extensive variation in a number of important biological traits, including coloration, venom production, and regeneration. Due to a lack of genomic tools, few genetic studies in squamates have been carried out. The leopard gecko, *Eublepharis macularius,* is a popular companion animal, and displays a variety of coloration patterns. We took advantage of a large breeding colony and used linkage analysis, synteny, and homozygosity mapping to investigate a spontaneous semi-dominant mutation, "Lemon Frost", that produces white coloration and causes skin tumors (iridophoroma). We localized the mutation to a single locus which contains a strong candidate gene, SPINT1, a tumor suppressor implicated in human skin cutaneous melanoma (SKCM) and over-proliferation of epithelial cells in mice and zebrafish. Our work establishes the leopard gecko as a tractable genetic system and suggests that a tumor suppressor in melanocytes in humans can also suppress tumor development in iridophores in lizards.

## Author summary

The squamates (lizards and snakes) comprise a diverse group of reptiles, with more than 10,000 described species that display extensive variation in a number of important biological traits, including coloration. In this manuscript, we used quantitative genetics and genomics to map the mutation underlying white coloration in the Lemon Frost morph of the common leopard gecko, *Eublepharis macularius*. Lemon Frost geckos have increased white body coloration with brightened yellow and orange areas. This morph also displays a high incidence of iridophoroma, a tumor of white-colored cells. We obtained phenotype information and DNA samples from geckos in a large breeding colony and used genome sequencing and genetic linkage analysis to localize the Lemon Frost mutation to a single

Whitney Foundation (to LG). Geckos Etc. Herpetoculture provided support in the form of salary for SS. The funders had no role in study design, data collection and analysis, decision to publish, or preparation of the manuscript.

**Competing interests:** Steve Sykes is the owner of Geckos Etc. Herpetoculture. There are no patents, products in development or marketed products associated with this research to declare, and this does not alter our adherence to PLOS policies on sharing data.

locus. This locus contains a strong candidate gene, SPINT1, a tumor suppressor implicated in human skin cutaneous melanoma. Together with other recent advances, our work brings reptiles into the modern genetics era.

## Introduction

Color-producing cells [1–5] contribute to animal coloration and patterns. Some cells, such as melanocytes, produce pigments chemically. Others, such as iridophores, produce colors structurally by making crystal platelets [6–9]. Iridophores are not present in mammals, but are widespread in insects, fish, birds, amphibians and reptiles. Different types of iridophores can lead to different colors, including blue [10,11], yellow [12], and white [13]. The size, morphology and organization of guanine crystals, which form reflective platelets within the iridophores, are considered the mechanisms of different colors [12,14–16]. The form, number and distribution of the iridophores determine coloration and patterning of the organism [11,17–20]. We know little of the molecular mechanisms of guanine crystal regulation [21,22]. In contrast with well-studied melanocytes, there have been few molecular genetic analyses involving iridophores. A recent study found that endothelin signaling regulates iridophore development and proliferation in zebrafish [23]. In mammals, this pathway is required for melanocyte development [24], suggesting that signaling pathways conserved in evolution can be adapted to regulate different types of chromatophores.

Many reptile species (*e.g.*, geckos, chameleons, snakes) are bred in captivity as companion animals, and breeders have established morphs with unique colors and patterns [2]. The inheritance of different color morphs is usually carefully documented by breeders. The common leopard gecko, *Eublepharis macularius*, is an especially attractive model to study the molecular regulation of coloration because dozens of color and pattern morphs have been established over the past 30 years of selective breeding. These morphs either intensify a particular color (S1A–S1I Fig) or rearrange coloration patterns (S1J–S1L Fig).

Uncontrolled proliferation of iridophores can lead to iridophoroma. White-colored iridophoroma is common in many reptile species [25], including green iguanas [26], captive snakes [27], bearded dragons [28] and veiled chameleons [29]. The genetic causes of iridophoroma in these species are unknown. Recently, histopathological findings of iridophoroma were reported in the Lemon Frost morph of leopard geckos [30]. This morph arose as a spontaneous mutation in a female hatchling from a cross between two wildtype leopard geckos. The mutation leads to increased white body coloration and brightened yellow and orange areas. The Lemon Frost color morph provides a unique resource for uncovering genetic regulation of iridophores and iridophoroma.

A draft leopard gecko genome assembly has been published, containing 2.02 Gb of sequence in 22,548 scaffolds, with 24,755 annotated protein-coding genes [31]. Embryonic development *in ovo* and blastema-based tail regeneration have also been staged and documented in great detail [32–34]. Here, we took advantage of these established resources and used quantitative genetics to gain insight into the molecular regulation of white color and iridophoroma in leopard geckos.

## Results

### The Lemon Frost allele is a spontaneous semidominant mutation

A male leopard gecko carrying the *lemon frost* (*lf*) allele, Mr. Frosty (Fig 1A and 1B), was crossed to 12 female leopard geckos of different genetic backgrounds. The F1 progeny, which

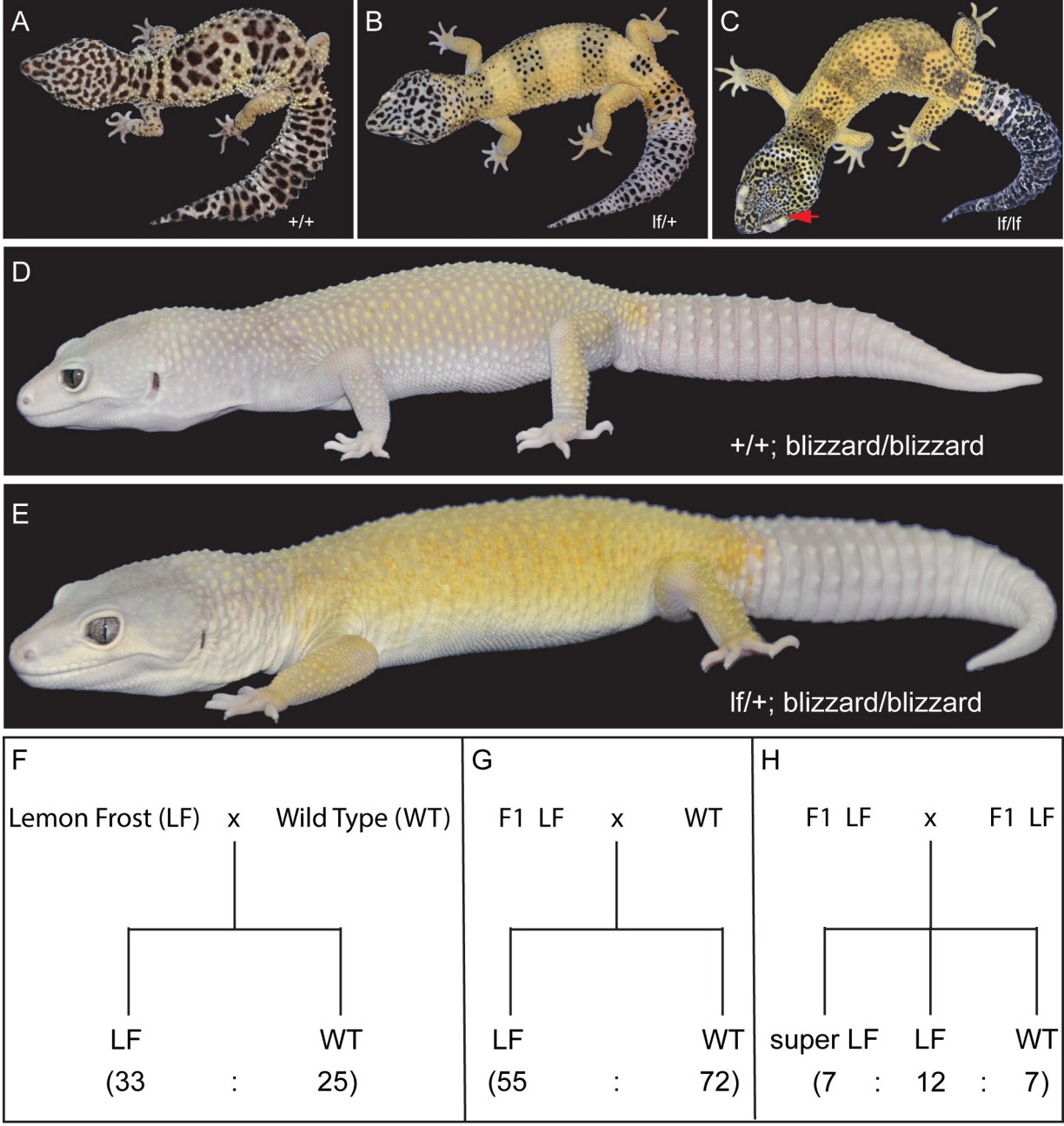

**Fig 1. The Lemon Frost mutant of the common leopard gecko, *Eublepharis macularius*.** (A) wild type; (B) heterozygous mutant; (C) homozygous mutant, with red arrow pointing to the eye lid; (D) blizzard mutant with minimal color; (E) Lemon Frost mutation (*lf*) on the blizzard background; (F-H) segregation of the *lf* allele. Lemon Frost (LF) denotes heterozygotes for the mutation; super LF denotes homozygotes for the mutation. All proportions are consistent with expectations for single-locus Mendelian inheritance (chi-square test p > 0.1).

were heterozygous for the *lf* allele, were backcrossed to the same maternal lines or intercrossed to establish a colony of more than 900 animals (S2 Fig). Homozygous F2 intercross progeny were named super Lemon Frost (Fig 1C). These homozygous mutants have an accentuated color phenotype and thickened skin, which is most apparent in their eyelids (Fig 1C, red arrow). Heterozygous Lemon Frost animals were also crossed to another mutant, Blizzard, which is light yellow without other colors or patterns (Fig 1D). The homozygous Blizzard

progeny carrying the *lf* allele displayed excessive white color in their heads and trunks, which brightened Blizzard's yellow color (Fig 1E). The *lf* allele also increased white color in the retina (Fig 1E). The segregation pattern of Lemon Frost in pedigrees is consistent with single-locus Mendelian inheritance (Fig 1F–1H). The *lf* allele is semidominant, as homozygous mutants have more pronounced phenotypes than do heterozygotes (Fig 1B, 1C and 1F–1H).

## The *Lemon Frost* allele leads to iridophoroma, with potential metastasis in homozygous animals

Three heterozygous Lemon Frost animals were recently reported to develop iridophoroma [30], a tumor of iridophores. Histopathological examination of the skin samples from homozygous mutants with accentuated phenotypes showed large solid sheaths of round to polygonal neoplastic cells that efface and expand the normal tissue architecture (S3 Fig). The cells have abundant cytoplasm with bright brownish intracytoplasmic pigment. The nuclei are eccentric and vary from round to fusiform. The white tumor masses stain dark with Hematoxylin and Eosin (H&E), and remain brightly reflective under dark-field illumination (S4A and S4B Fig), consistent with their nature as iridophores [10,35–38]. Imaging with Transmission Electron Microscopy (TEM) showed that the *lf* allele led to both increased numbers of neoplastic iridophores and increased production of reflective platelets within each iridophore [22] (S4C Fig). In addition to skin, other affected organs in homozygous mutants include liver, eye, and muscle. The interpretation of the widespread neoplastic nodules is that the tumors are malignant iridophoroma. The increased number of iridophores and increased production of reflective platelets within the iridophores are the likely mechanisms of increased white color in the Lemon Frost morph.

More than 80% of both male and female animals carrying the *lf* allele developed white tumors 6 months to 5 years after birth. The tumors manifest as patches of white cells in the skin, which are most evident on the ventral side of the animal (Fig 2A). The tumor skin can be severely thickened and leathery (Figs 2B and S3). It is resistant to liquid nitrogen freezing, or to Dounce homogenization, making RNA extraction infeasible. In severe cases in heterozygous mutants, the tumors develop into skin protrusions (Fig 2C, left), which contain dense white masses (Fig 2C, right). Tumors cover a greater fraction of the skin of homozygous mutants. Surprisingly, these tumors rarely develop into skin protrusions as in heterozygous animals. Instead, they manifest as well-demarcated, white, thickened patches on the ventral skin (Fig 2A), thickened layers of white masses all over the dorsal skin (Fig 2B), white, multifocal, variably sized, well-demarcated nodules in the liver, and patches of white cells in the oral cavity (Fig 2D).

## Linkage and association analysis in a breeding pedigree

To identify the genetic locus that regulates white color and tumor growth in Lemon Frost mutants, we used restriction site-associated DNA sequencing (RAD-Seq) to genotype 188 animals from the breeding pedigree (Figs 3 and S2), including 33 super Lemon Frost (*lf/lf*), 116 Lemon Frost (*lf/+*), and 39 wild-type (+/+) individuals. We identified a total of 14,857 variants covering 2,595 scaffolds of the genome assembly. To map the Lemon Frost locus, we tested the effect of allelic dosage at each marker on white coloration of the geckos in a standard semidominant association mapping framework, accounting for population structure through the use of marker-based relatedness. We used a p-value threshold of 7.09e-5 (Methods) to control the false positive rate at 1%. Forty-eight markers on 31 scaffolds were significantly associated with white coloration (S1 Table). The top two association signals corresponded to scaffolds 6052 and 996.

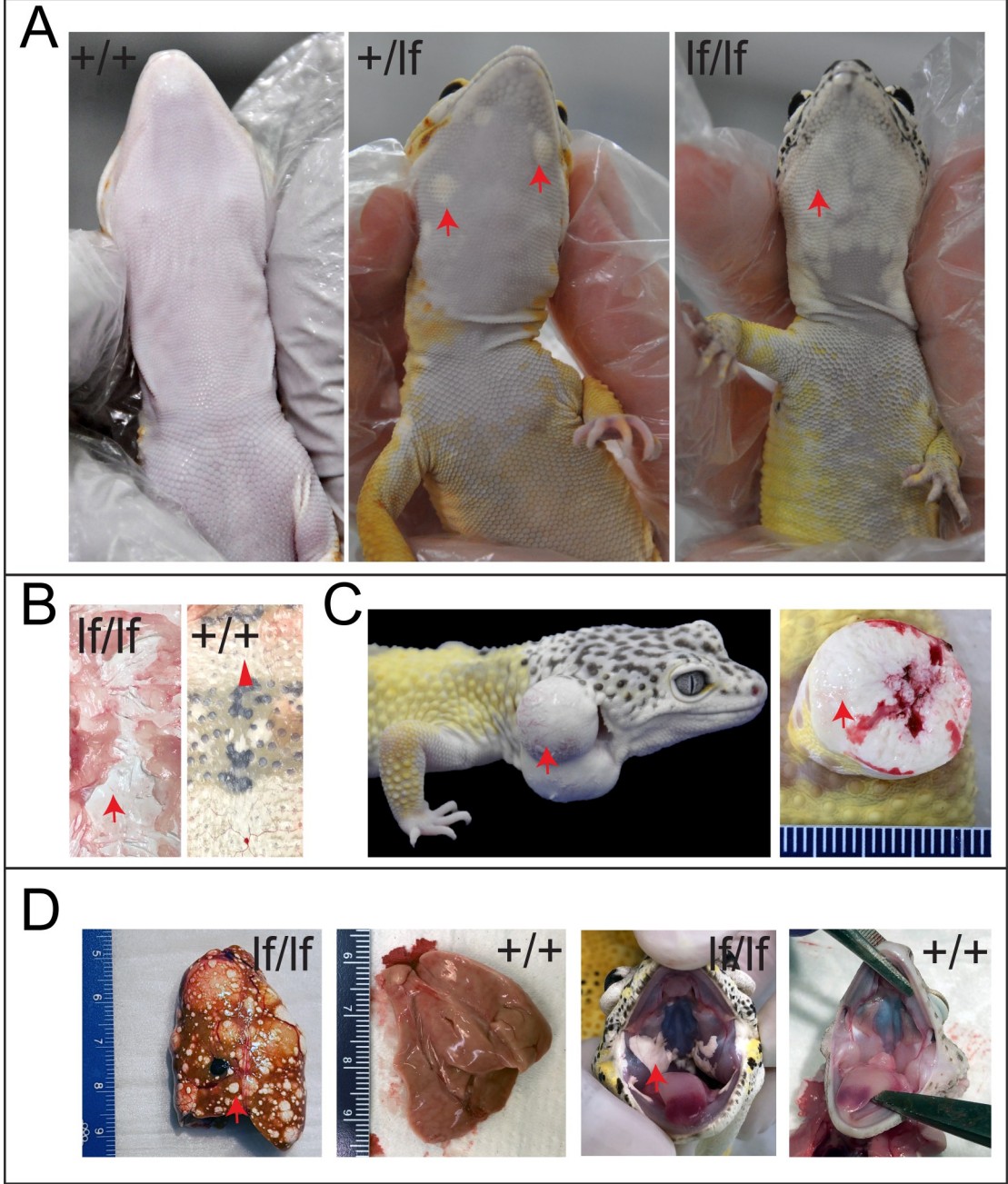

**Fig 2. Tumor growth and metastasis in the Lemon Frost mutant.** Designations are homozygous mutant (*lf/lf*); heterozygous mutant (*lf/+*); wild type (+/+). (A) tumors in ventral skin; (B) thick layers of white tumor cells (*lf/lf*) vs. normal white cells (+/+); (C) outgrowth of white tumor cells (*lf/+*); (D) metastasis of white tumor cells in the liver and oral cavity. Red arrows: white colored tumor cells. Arrowhead in B: normal white cells.

## Synteny analysis and homozygosity mapping

Because the gecko genome assembly is highly fragmented, we used synteny to examine whether the 31 scaffolds associated with coloration belong to a single genomic interval. We compared the gecko scaffolds to homologous regions of several vertebrate species with chromosome-scale genome assemblies: green anole [39], chicken [40] and human [41]. We

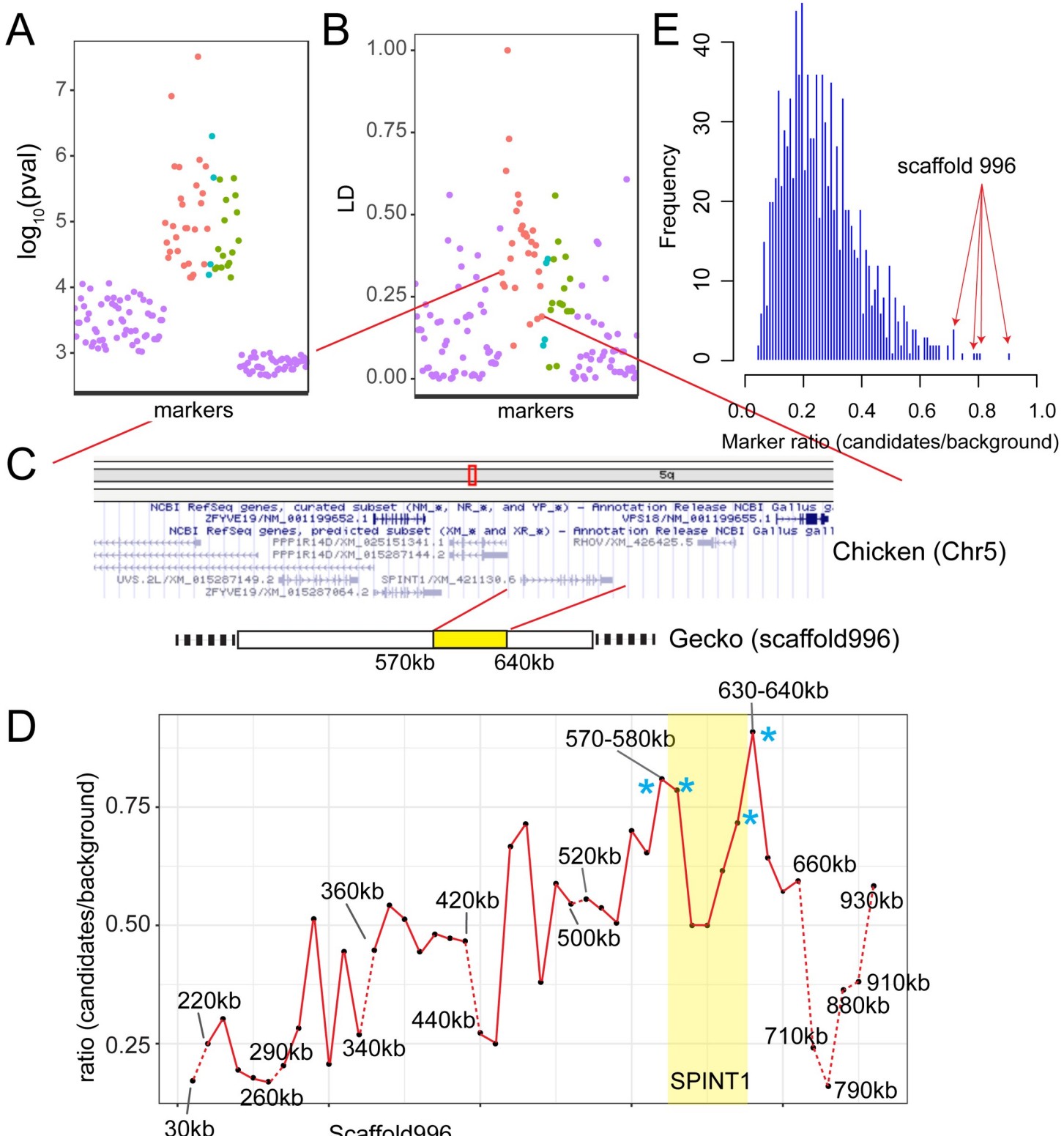

**Fig 3. Localization of the Lemon Frost mutation.** (A) p-value for association with white color and (B) linkage disequilibrium for 28 markers syntenic to chicken chromosome 5 (red, ordered by synteny), 4 markers syntenic to chromosome 7 (cyan), and 16 markers without synteny information (green). Randomly selected markers that are not associated with white color (purple). (C) A schematic of the region showing synteny and gene annotation. (D) Fraction of markers showing expected allele frequency pattern in pools, plotted for 10kb windows along scaffold 996. The four windows with the highest fraction are marked by asterisks and span the location of the

gene SPINT1. Windows with fewer than 5 variants were not plotted (dashed red lines). (E) Genome-wide distribution of the fraction of markers showing expected allele frequency pattern in pools for all 10 kb windows. The 4 highest windows on scaffold 996 (red arrows) marked in D are among the 6 highest windows in the entire genome.

found that 17 of the 22 scaffolds that have synteny information (including scaffolds 6052 and 996) correspond to one region on chicken chromosome 5, human chromosome 15, and green anole chromosome 1 (Fig 3A–3C and S1 Table). Three additional scaffolds also have synteny to the green anole chromosome 1. The remaining two scaffolds without synteny to the green anole chromosome 1 were more weakly associated with coloration. The 28 markers on these 17 scaffolds are in linkage disequilibrium (Fig 3B), which decays with distance when markers are ordered by synteny (Fig 3B). These results indicate that a single genomic region is associated with the Lemon Frost phenotype, as expected for a new mutation with a Mendelian segregation pattern.

To narrow down the location of the causal gene within this genomic region, we used whole genome sequencing and homozygosity mapping. We pooled DNA from 25 super Lemon Frost genomes (*lf/lf*), 63 Lemon Frost genomes (*lf/+*), and 71 wildtype geckos (+/+) and sequenced each pool to 30x coverage. We reasoned that the *lf* mutation in Mr. Frosty and its flanking variants should form a haplotype that would be found in the super Lemon Frost pool with 100% frequency, in the Lemon Frost pool with 50% frequency, and would not be seen in the wildtype pool. We scanned the genome in 10 kb windows and measured the fraction of heterozygous variants from Mr. Frosty that followed this expected pattern in the pools (S2 Table). This statistic was highest for a window on scaffold 996 (S2 Table and Methods), the main candidate scaffold from statistical mapping (S1 Table). The expected frequency pattern was observed for 20 of 22 variants in this window (630-640kb on scaffold 996). Four of the top six intervals fall in the region from 570kb to 640kb on scaffold 996, with the signal decaying with distance away from this region (Fig 3D and 3E). The linkage between this region and Lemon Frost was replicated in an independent 3-generation backcross between Mr. Frosty and a Sunburst Tangerine morph (Fig 4 and S3 Table). These results indicate that scaffold 996 contains the Lemon Frost mutation.

## SPINT1 is a strong candidate gene for the Lemon Frost phenotype

The genomic interval spanning positions 570kb-640kb on scaffold 996 contains a single gene, SPINT1. SPINT1 (serine peptidase inhibitor, Kunitz type 1), also known as hepatocyte growth factor activator inhibitor type 1 (HAI-1), is a transmembrane serine protease inhibitor expressed mainly in epithelial cells [42–44]. It is the only gene in the larger associated region reported to be a suppressor of epithelial cell tumors in model organisms and in humans [42,45–56]. Because the breeding and transmission data indicate that the *lf* allele arose from a single spontaneous mutation, we reasoned that a mutation disrupting SPINT1 causes the overproliferation of white-colored skin cells in Lemon Frost geckos.

The Lemon Frost SPINT1 allele differs from the reference genome assembly at two positions in the exons, as well as at 147 positions in the introns and the 3'UTR (S4 Table). This large number of variants is a consequence of differences in genetic background between Mr. Frosty's parents and the non-Lemon Frost individual used to generate the reference, and makes it challenging to identify the causal mutation. Both differences in the coding sequence of SPINT1 are synonymous. Notable differences in non-coding regions include 7 large insertion/deletions (indels) in the introns and a 13-nucleotide insertion in the 3'UTR (CAAGTGTATGTAT). Indels in introns and promoters of SPINT1 have been reported to lead to loss of SPINT1 function in fish and mice [51,52,56].

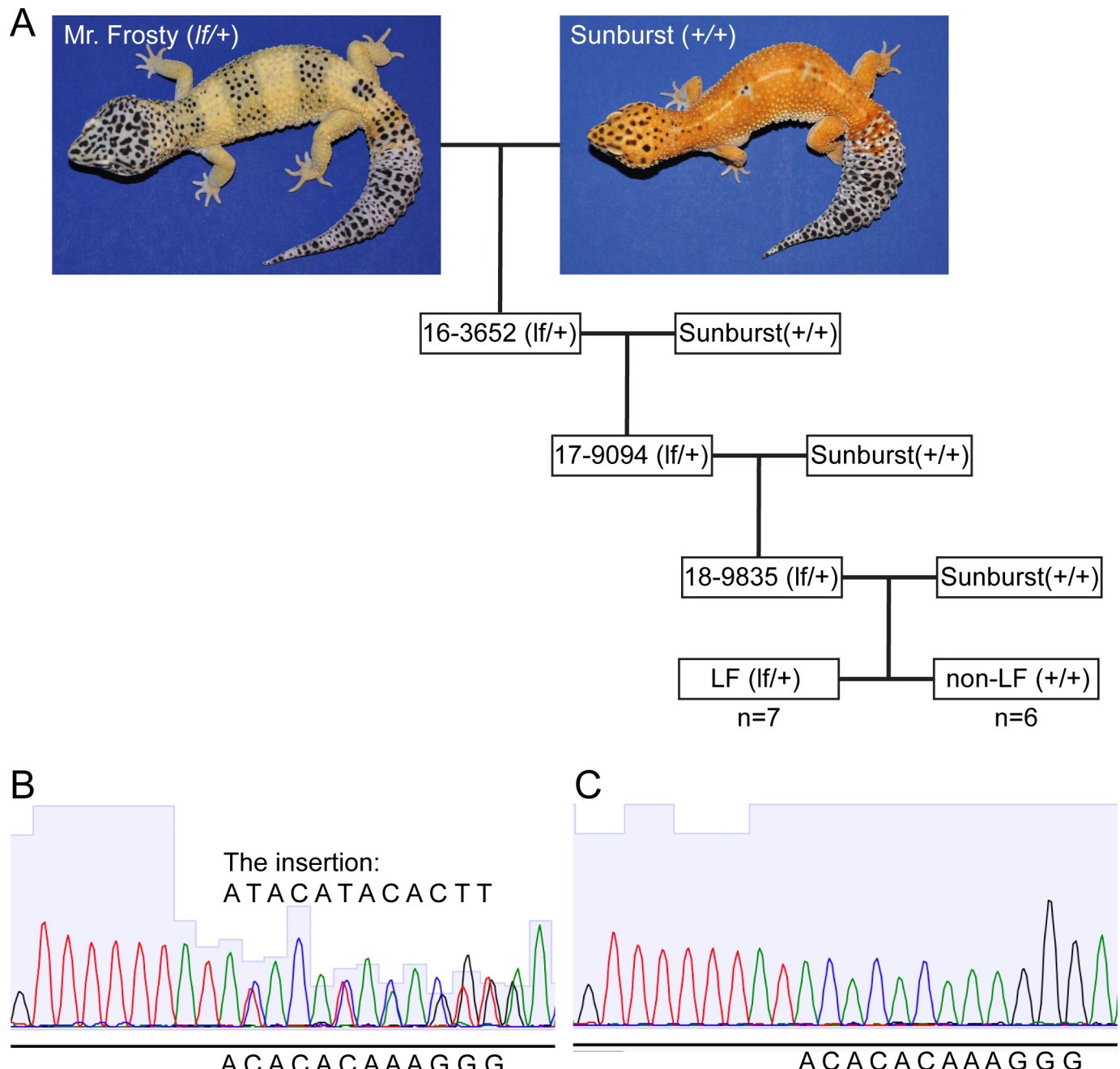

**Fig 4. The *lemon frost* allele in a backcross.** (A) We genotyped 7 progeny with the Lemon Frost phenotype and 6 wild type progeny from the third generation of a backcross of Mr. Frosty to the Sunburst line for markers in the SPINT1 region and observed a consistent inheritance pattern. (B) Sequencing chromatogram of a heterozygous animal (*lf*/+) at an insertion marker. (C) Sequencing chromatogram of a homozygous animal (+/+) at the same insertion marker.

We used BLASTn to examine whether some of the non-coding differences between the reference and the Lemon Frost SPINT1 alleles are located in well-conserved vertebrate sequences. We found that intron 1 has >90% sequence identity to Gekko japonicus and >80% sequence identity to Zootoca vivipara. Intron 8 has >80% sequence identity to Saurodactylus brosseti. The 3' UTR has 79% identity to Gekko japonicus. We also found that human SPINT1 introns 1/2 and the 3' UTR are the most conserved non-coding regions of the gene among 100

vertebrates by PhyloP analysis [57–59], with conservation scores that are comparable to those for exons 8 and 10. The ENCODE candidate cis-regulatory elements (cCRE) analysis showed that introns 1/2 and the 3' UTR are the major regions with multiple candidate enhancers [57,60]. Similar results were obtained for the mouse SPINT1 gene. These observations suggest that some of the 22 sequence variants in intron 1 or 20 variants in the 3' UTR of the leopard gecko SPINT1 may be functionally important.

Sequencing of RNA extracted from normal gecko skin and from skin peripheral to tumors in homozygous mutants confirmed that SPINT1 is expressed in this tissue (S5 Fig). However, we did not observe a significant difference between homozygous mutants and wildtype geckos in SPINT1 mRNA levels or splicing patterns. This result suggests that the putative causal mutation in SPINT1 may alter translation or protein activity, rather than transcription. Alternatively, the mutation might reduce SPINT1 expression only in tumors, which are refractory to RNA extraction as noted above.

## Discussion

Several lines of evidence support our hypothesis that a defect in SPINT1 causes iridophoroma in Lemon Frost geckos. First, SPINT1 function is dosage-dependent, consistent with our observation that Lemon Frost is a semi-dominant phenotype. In humans, carcinoma tissues *in vivo* and carcinoma-derived cell lines *in vitro* have reduced SPINT1 on the cell membrane [61,62] through enhanced shedding of the extracellular domain or decreased mRNA or protein expression. Reduced expression of SPINT1 has been associated with a negative prognosis of human Skin Cutaneous Melanoma (SKCM) [45] and pancreatic ductal adenocarcinoma [46]. Knockdown of SPINT1 expression by siRNA in cancer cell lines led to increased invasion or metastasis [47,62,63]. Second, loss of SPINT1 function in fish and mice leads to tumor formation in epithelial cells. In mice, homozygous deletion of SPINT1 leads to disrupted placental basement membranes and embryonic lethality [52,54]. Rescued mosaic animals developed scaly skin with hyperkeratinization [55]. Intestine-specific deletion of SPINT1 leads to increased tumor growth of intestine epithelium [48]. Increased expression of SPINT1 in the skin abrogated matriptase-induced spontaneous skin squamous cell carcinoma [64]. In zebrafish, reduced expression led to hyperproliferation of basal keratinocytes [51] and enhanced proliferation of epithelial cells [56]. Furthermore, SPINT1 deficiency was used to establish a disease model for Skin Cutaneous Melanoma (SKCM) in zebrafish [45]. In all three studies in zebrafish, skin inflammation was observed. Third, insertions in introns [51,52] and promoters [56] have caused loss of SPINT1 function. Together with our genetic localization of the *lf* locus to SPINT1, these lines of evidence make this gene a very strong candidate for the Lemon Frost phenotype.

Molecular genetics in reptiles is not well established due to long reproductive cycles and challenges in laboratory breeding. Early work focused on careful documentation of patterns of inheritance [2,65]. Molecular studies have examined sequence variants in a candidate pigmentation gene, melanocortin-1 receptor, and their association with melanic or blanched phenotypes in different species and ecological niches [66–73]. Recent work in the wall lizard [74] and the corn snake [75] used a similar sequencing-based approach to our study to identify the molecular basis of coloration polymorphisms in these species. Successful use of CRISPR-Cas9-mediated gene editing to mutate the tyrosinase gene has been reported in the lizard *Anolis sagrei* [76]. Although this species is only distantly related to the leopard gecko, this advance offers promise that targeted studies of the role of SPINT1 mutations in the Lemon Frost phenotype will become possible.

Most of our knowledge about molecular and cellular regulation of iridophores derives from work in zebrafish [11,13,17,19,23,77–85]. Interestingly, few cases of iridophoroma have been reported in zebrafish [86]. Our data suggest that an evolutionarily conserved gene, SPINT1, regulates the proliferation of white iridophores in the leopard gecko. Increased production of reflective platelets in Lemon Frost iridophores is a novel finding that warrants further investigation, for instance by manipulating SPINT1 expression. The tumor suppressor function of SPINT1 establishes a potential link between iridophoroma and regulation of white coloration in reptiles. Our work suggests that cancer genes can play as important a role in iridophores as they do in melanocytes and melanoma [87], and that Lemon Frost leopard geckos may serve as a disease model to study Skin Cutaneous Melanoma.

## Methods

### Ethics statement

All activities involving animals included in this manuscript were approved by the University of California, Los Angeles (UCLA) Institutional Animal Care and Use Committee, approval number: ARC #2018–035 (6/19/2018~6/18/2021).

### Gecko maintenance and experimental procedures

Leopard geckos were acquired from a commercial breeder. Housing conditions at UCLA included: room temperature of 70–80 F, cage temperature of 72–95 F, room relative humidity between 30–60%, and a 12:12 hours light cycle. A heating pad was provided at one side of the cage to establish a temperature gradient. Animals were singly housed in polycarbonate cages with cardboard lines (Techboard) at the bottom, water was provided in bowls inside the cage, and PVC pipe pieces and plastic plants were offered as environmental enrichment. Geckos were fed 2–6 fresh crickets and 2–4 mealworms three times per week.

Geckos were euthanized with an intracoelomic injection of sodium pentobarbital (Euthasol) at a dose of 100–200 mg/Kg. Immediately after euthanasia, a necropsy was performed, including external examination, body and organ weighing, gross assessment of normal and abnormal tissues, and tissue collection for histopathology processing and assessment. Normal and abnormal tissues were fixed in 10% formalin, embedded in paraffin, sectioned, and stained with H&E for pathologic evaluation.

### Phenotyping

Lemon Frost and super Lemon Frost phenotypes were scored by experienced breeders who used visual inspection to determine an increase in white color of the body, eye, and belly compared to normal wildtype animals. The Lemon Frost mutation increases the white base color of the gecko over its entire body. This results in an overall brightening of white, yellow, and orange colors of the gecko. The effects of the mutation can also be observed in a whitening of the eyes and in white color spreading down the sides onto the belly. Pictures were taken for each animal to document the phenotypes.

### Genotyping

Genomic DNA was extracted from fresh tail tips with Easy-DNA gDNA purification kit (K180001, ThermoFisher), or from the saliva with PERFORMAgene (PG-100, DNAgenotek). Genomic DNA extracted from saliva was further purified with ethanol precipitation before genotyping assays. DNA libraries for whole genome sequencing were prepared with Nextera DNA Library Prep Kit (FC-121-1031, Illumina). Libraries for RADseq were prepared

according to the procedures of Adapterama III [88] with few modifications. Libraries were sequenced on a HiSeq 3000 (Illumina).

Only scaffolds larger than 5kb in the draft genome assembly were used as a reference. RAD-seq reads and Whole Genome Sequencing (WGS) reads were aligned to the leopard gecko draft genome [31] with bwa mem (version 0.7.17) [89]. Variants for WGS were identified with GATK (version 4.1.4.1) [90]. Variants for RADseq were identified with Stacks (version 2.41) [91,92]. All variants were filtered with VCFtools (version 0.1.14) [93]. First, variants had to be are bi-allelic and heterozygous in Mr. Frosty. The ratio of the reads for the two alleles was required to be in the range 0.4–0.6. Second, only high-quality variants were used in homozygosity mapping or statistical mapping (DP> = 30, GQ> = 30).

## Transcriptome sequencing

Skin tissue samples around 6mm in diameter were taken from the ventral side of the geckos after anesthetization with 1–5% isoflurane. As tumor tissues are refractory to RNA extraction, flanking tumor-free tissue samples were taken for homozygous Lemon Frost animals. All samples were homogenized with TissueRuptor in buffer RLT immediately after collection. Lysates were immediately frozen on dry ice until all tissues were collected from animals. Then all lysates were centrifuged for 5 minutes at 13,000 rpm to remove debris. Supernatants were taken to fresh tubes, and mRNA was extracted according to the procedures of RNeasy Fibrous Tissue Mini Kit (74704, QIAGEN).

Libraries of extracted mRNA were prepared with RNA HyperPrep kit (KAPA) and sequenced on a HiSeq 3000 (Illumina). RNA-seq reads were mapped to the leopard gecko draft genome [31] using HISAT2 with default parameters. Identification of alternative and differential splicing events was performed using JuncBase [94]. Gene expression was compared using Sleuth [95] after RNA transcript abundance was quantified using Kallisto [96].

## Pathology

Complete postmortem examination was performed, and representative tissue samples were obtained. All tissues obtained at necropsy were preserved in 10% neutral-buffered formalin solution for up to 5 days before being processed and embedded in paraffin. All tissues were sectioned at 5 μm, and routinely stained with Hematoxylin and Eosin.

## Statistical mapping

Biallelic markers with minor allele frequency of less than 5% and with fewer than 10 individuals called as homozygous for both the reference and alternative alleles were excluded from mapping and kinship matrix construction. A kinship matrix was calculated using the function *A.mat* with default parameters from the rrBLUP [97] R package. Phenotype was encoded as 0 for wild type, 1 for Lemon Frost, and 2 for super Lemon Frost. Association statistics between this phenotype vector and marker genotypes were computed using the function *gwas2* in the NAM [98] R package using a linear mixed model with a random effect of kinship to control for population structure. The effective number of tests was computed to be 141.1 based on the procedure of Galwey et al [99]. A family-wise error rate significance threshold was calculated as 0.01/141.1 or p<7.09e-5.

## Homozygosity mapping

Pooled animals and Mr. Frosty were sequenced to ~30x coverage on a HiSeq 3000 (Illumina). Variants were identified with GATK and filtered with VCFtools. Biallelic heterozygous

variants from Mr. Frosty, including indels, were used as markers to localize the Lemon Frost mutation. Allele ratios (AF) were calculated by dividing the read count of alternative alleles by the sum of the counts of reference alleles and alternative alleles. Variants closely linked to the Lemon Frost mutation are expected to have AF between 0.4 and 0.6 in the Lemon Frost pool and in Mr. Frosty, AF > 0.85 in the super Lemon Frost pool, and AF < 0.15 in the wildtype pool. The number of variants meeting these criteria was counted for every 10kb genome interval. The fraction of such variants among all variants heterozygous in Mr. Frosty within the interval was then calculated. Intervals with fewer than 5 variants were excluded because they could not provide statistically meaningful results.

### Transmission electron microscopy

Dissected skin tissues were fixed in 2.5% glutaraldehyde and 4% formaldehyde in 0.1 M sodium cacodylate buffer overnight at 4˚C. After being washed in PBS, samples were post-fixed in 1% osmium tetroxide in 0.1M sodium cacodylate, and dehydrated through a graded series of ethanol concentrations. After infiltration with Eponate 12 resin, the samples were embedded in fresh Eponate 12 resin and polymerized at 60˚C for 48 hours. Ultrathin sections of 70 nm thickness were prepared, placed on formvar-coated copper grids, and stained with uranyl acetate and Reynolds' lead citrate. The grids were examined using a JEOL 100CX transmission electron microscope at 60 kV, and images were captured by an AMT digital camera (Advanced Microscopy Techniques Corporation, model XR611).

### Supporting information

**S1 Fig. Coloration and pattern diversity of the common leopard gecko, *Eublepharis macularius*.** (A) wild type; (B) black night; (C) variant of black night; (D) granite snow; (E) gem snow; (F) white knight; (G) sunburst tangerine; (H-I) variants of sunburst tangerine; (J) red stripes; (K) bold stripes; (L) rainbow.
(TIF)

**S2 Fig. Breeding pedigree of the Lemon Frost mutation.** Mr. Frosty, the original carrier of the spontaneous Lemon Frost mutation, was bred to 12 female geckos from different genetic backgrounds. F1s carrying the *lf* allele were bred among themselves or back to their female parent, producing the second generation of animals heterozygous or homozygous for the *lf* allele. Blue: *lf/lf*; green: *lf/+*; red: *+/+*. Dashed line: same individual/line.
(TIF)

**S3 Fig. Histopathology of skin tumors.** (A) Thick layers of white tumor tissue (star) infiltrating white skin (arrow). (B) Skin biopsies organized and fixed in a paper roll for sectioning. (C) H&E staining of the skin sections. Arrow: skin; star: infiltrated tumor mass. (D) H&E staining of the skin sections showing normal skin cells and neoplastic cells (star). Neoplastic cells have eccentric and condensed nuclei.
(TIF)

**S4 Fig. Potential metastasis of iridophoroma.** (A) In normal skin, cell nuclei are oval and perpendicular to the skin surface. In Lemon Frost skin, cell nuclei are flat, elongated and parallel to the skin, reminiscent of epithelial-to-mesenchymal transition. (B) Iridophoroma in the liver, stained dark in H&E sections. In dark field imaging, iridophores are bright white. Such iridophores invade blood vessels in the tissue (red arrows). (C) In TEM imaging, white tumor skins in super LF are filled with abundant iridophores with excessive brightly reflective crystals (Tumor). In normal skin, iridophores are much fewer and have less crystals (Normal).
(TIF)

**S5 Fig. SPINT1 expression in gecko skin.** SPINT1 mRNA reads from transcriptome sequencing were aligned to the genome and visualized in IGV. Top 3 rows show samples from homozygous mutants. Bottom 3 rows show samples from wild type geckos. Skin tissue adjacent to the tumors was used in the mutants. Peaks mark SPINT1 exons. The last exon on the right is transcribed together with the 3'UTR.
(TIF)

**S1 Table. Scaffolds associated with Lemon Frost phenotype.**
(XLSX)

**S2 Table. Candidate Lemon Frost mutations on SPINT1.**
(XLSX)

**S3 Table. Genotyping results of the backcross.**
(XLSX)

**S4 Table. Candidate and background mutations in homozygosity mapping.**
(XLSX)

## Acknowledgments

We thank Aaron Miller, Jasmine Gonzalez, Kendall Placido and James Walter for their assistance in gecko DNA collection and phenotyping. We thank members of the Kruglyak lab for helpful feedbacks on the project, and Giancarlo Bruni, Stefan Zdraljevic, Eyal Ben-David and Olga Schubert for helpful comments on the manuscript. We thank Chunni Zhu (Electron Microscopy Core Facility, UCLA Brain Research Institute) for her assistance in TEM sample processing and imaging. We thank Jonathan Eggenschwiler and Douglas Menke for helpful discussions.

## Author Contributions

**Conceptualization:** Longhua Guo, Leonid Kruglyak.

**Data curation:** Longhua Guo, Zain Kashif, Elise Pham, Katarina Ho.

**Formal analysis:** Longhua Guo, Joshua Bloom, Elaine Huang, Xinshu Grace Xiao.

**Funding acquisition:** Longhua Guo, Leonid Kruglyak.

**Investigation:** Longhua Guo, Joshua Bloom, Steve Sykes, Ana Alcaraz, Sandra Duarte-Vogel, Leonid Kruglyak.

**Methodology:** Longhua Guo, Joshua Bloom, Ana Alcaraz, Sandra Duarte-Vogel, Leonid Kruglyak.

**Project administration:** Longhua Guo, Leonid Kruglyak.

**Resources:** Steve Sykes.

**Software:** Joshua Bloom, Elaine Huang.

**Supervision:** Xinshu Grace Xiao, Leonid Kruglyak.

**Validation:** Longhua Guo, Leonid Kruglyak.

**Visualization:** Longhua Guo, Leonid Kruglyak.

**Writing – original draft:** Longhua Guo, Leonid Kruglyak.

**Writing – review & editing:** Longhua Guo, Zain Kashif, Katarina Ho, Ana Alcaraz, Sandra Duarte-Vogel, Leonid Kruglyak.

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
