## [Decision Letter · Decision Letter 0]

7 Mar 2021

Dear Leonid,

Thank you very much for submitting your Research Article entitled 'Genetics of white color and iridophoroma in “Lemon Frost” leopard geckos' to PLOS Genetics.

The manuscript was fully evaluated at the editorial level and by two independent peer reviewers. The reviewers appreciated the attention to an important topic but identified some minor concerns that we ask you address in a revised manuscript

We therefore ask you to modify the manuscript according to the review recommendations. Your revisions should address the specific points made by each reviewer.

[LINK]

Yours sincerely,

Hopi E. Hoekstra, PhD

Consulting Editor - PLoS Genetics

PLOS Genetics

Hua Tang

Section Editor: Natural Variation

PLOS Genetics

Thank you for your submission to PLoS Genetics. As you will see, your manuscript was reviewed by two external experts, both of whom were very positive about the work but requested that some minor revisions are made. I suspect the changes and additional analyses will require a minimum amount of work, and your manuscript will likely not need to be reviewed externally again.

Both reviewers found that this work was significant given few mapping studies have been conducted in lizards and they found the approach to be rigorous. Both also are convinced that the SPINT1 gene is a strong candidate gene, and agree that functional validation, which would be important in a model system, is unreasonable to ask in a non-model system. However, given there is no functional validation and little new experimental evidence (beyond a strong statistical association), both reviewers ask that the language about the role of SPINT1 be toned down (e.g. Line 214-215). In addition, Reviewer 2 suggests one additional analyses to examine conservation of the non-coding sequence variants you identified in Lemon Frost to determine if any are likely be functionally important. This seems quite straightforward, and would help round out your results. Finally, Reviewer 1 makes several useful suggestions to improve the structure of the paper, which will likely better highlight and broaden the appeal of your work. 

Congratulations on a nice piece of work; we look forward to seeing a revised version.  -Hopi

Reviewer's Responses to Questions

**Comments to the Authors:**

Reviewer #1: Using a combination of techniques, Guo et al. map the genetic basis of the Lemon Frost phenotype, which occurred spontaneously in a Leopard Gecko, to a small genomic region. This region contains only one gene (SPINT1), and, strikingly, this gene makes for an excellent candidate gene, given its known function as a tumor suppressor gene and the high incidence of iridophoromas in individuals carrying the Lemon Frost allele.

I think the authors’ main conclusions that a variant in SPINT1 likely underlies the LF phenotype is supported by their data and, overall, I very much enjoyed reading this manuscript. I congratulate the authors on their work. That being said, I think the manuscript would benefit from a bit of re-structuring (e.g. previous results presented in results and not introduction) and it seems a bit unbalanced in regard to its two central themes of “white coloration” and “iridophoroma” (see below). There are also a few additional points that the authors should/might want to address (e.g. synteny analysis).

Comments:

- I really appreciate that the authors made an effort to keep the introduction short and concise. That being said, I think there is an opportunity for the authors to flesh out the main motivation to investigate the genetic basis of coloration (especially iridophore-based) a bit more. The authors are already doing this to some extent, but, at the moment, the motivation of “There have been few molecular genetic analyses of the regulation of chromatophores in cells other than melanocytes” (ll. 41-42) could maybe be strengthened to draw readers’ attention. Consider adding one or a few sentences to expand on what can be learned from studying coloration and iridophores (i.e. expanding on the points in ll. 42-45 and more generally what is and what is not known). Also, and maybe more importantly, there is no mention of the link between the Lemon Frost phenotype and iridophoromas in the entire introduction (although it is mentioned in the abstract). Given that this is a major part of the results/manuscript, I think the manuscript would greatly benefit from a general paragraph on iridophoromas in the introduction.

- Related to my point above, at the same time, essentially the entire discussion seems to be about iridophoromas and hardly touches upon the genetic basis of white coloration at all. I understand that the two are intricately linked for this phenotype, but I really think that there could be more of a balance of the two themes (coloration and iridophoromas). For example, the Lemon Frost phenotype exhibits a striking banding pattern, but this is hardly (at all) discussed in the entire manuscript.

- ll. 63-66: “A spontaneous mutation occurred in a female hatchling from a cross between two wildtype leopard geckos. This mutation increased the white color of the leopard gecko, resulting in brightened white and yellow colors. This unique color morph was named Lemon Frost(19)”.

-> I recommend moving this information to the introduction. It seems a bit misplaced in the results section, since the Lemon Frost phenotype apparently first occurred around 2015 and has been described before (reference 19). I think moving these sentences would help readers to understand what is new to this study and what was known before.

- ll. 81-95: Similar to above, it seems that the authors are at least to some extent reporting previously published results here (references 19 and 24). I think it would be helpful if the authors tried to make it more explicit what exactly was done in this study and how their new data add to previously published results (e.g. do the authors provide new histological details specific to homozygous mutants, more quantitative data, etc.)

- ll. 91-92: “Microscopy (TEM) showed that the lf allele led to both increased numbers of neoplastic iridophores and increased production of reflective platelets within each iridophore”

-> Does this (increased number of iridophores and increased production of platelets) tell us anything about the mechanism and is it consistent with the function of SPINT1? Unless I missed it, it seems that the increased production of reflective platelets is not mentioned in the discussion at all. Maybe the authors can come back to this result in the discussion.

- ll. 127-130: “We found that 17 out of 22 scaffolds that have synteny information (including scaffolds 6052 and 996) correspond to one region on chicken chromosome 5 and human chromosome 15 (Figure 129 3A-C, Supplemental Table).”

-> Do the authors have any explanation for the other five scaffolds? It seems that four of them are in synteny with chr. 7 in chicken. Given the inheritance pattern, I think the main conclusion is probably robust and I highly doubt that there is a second fl locus. It is probably more likely that this due to a chromosomal rearrangement. Nonetheless, given that this block of four scaffolds includes scaffold707, which shows the third highest association signal (Supplementary Table), I think the authors need to mention/discuss this further. It might even be worth it for the authors to expand their synteny analyses to other reference genomes (e.g. Anolis carolinensis or the recently published high-quality Goodes thornscrub tortoise genome).

- ll. 214-215: “We found that an evolutionarily conserved gene, SPINT1, regulates the proliferation of white iridophores in the leopard gecko.”

-> Given that the authors have so far only established association, this statement seems too strong (i.e. it implies functional validation) and should be toned down.

- ll. 244-246: “Lemon Frost and super Lemon Frost phenotypes were determined according to a list of rules, based on increased white color of the body, eye, and belly compared to normal wildtype animals (http://www.geckosetc.com/lemon_frost_info.html).”

-> Please note that the link does not work (returns “The page you requested could not be found.“). In any case, I think it would be important if the authors described their phenotyping approach in the manuscript rather than just referring to an external source/website.

- Methods: Please provide version numbers for all software (e.g. GATK, vcftools, etc.)

- l. 262: “All variants were filtered with VCFtools.” and l. 305 “Variants were identified with GATK and filtered with VCFtools.”

-> Filtered how? Please provide details. Is this is described somewhere else, please add “see below” or something alike to help readers find this information.

- Supplementary table

-> The supplementary table is missing any sort of annotation. Please add headers inside the table for each sheet, explaining abbreviations, etc.

Reviewer #2: In this study Guo and colleagues investigate the genetic basis of Lemon Frost, a Leopard gecko mutation that results in a white color phenotype and skin tumors. Breeding experiments demonstrate a semi-dominant pattern of inheritance from a single locus. By performing RAD-seq on individuals from a breeding pedigree, Guo et al. localize Lemon Frost to region of the gecko genome that is syntenic to chicken chromosome 5 and human chromosome 15. Whole-genome sequencing of pooled DNAs from wt, heterozygous and homozygous individuals enabled the authors to further localize the mutation to region that contains the SPINT1 gene. However, no nonsynonymous coding mutations were found in SPINT1 and transcriptome analyses of homozygous mutantskin did not reveal changes in SPINT1 expression level or splicing. Nevertheless, functional studies of SPINT1 in zebrafish and the association of SPINT1 deficiency with Skin Cutaneous Melanoma (SKCM) in humans makes SPINT1 an extremely compelling candidate gene for Lemon Frost. The study is well designed and the authors’ statement in the abstract that SPINT1 is a “strong candidate gene” is generally well-supported. As the authors point out, very little molecular genetic research has been performed in reptiles. Therefore, it is exciting this work on a gecko model that may be relevant for understanding the biology of SKCM. Although functional validation of the candidate gene in geckos is currently not practical, I would still like to see some additional evidence that mutation of SPINT1 is responsible for the Lemon Frost phenotype.

Comments

1) Are any of the sequence differences found on the Lemon Frost allele of SPINT1 in well-conserved sequences? Is there evidence that any of the non-coding differences are in regions that are functionally conserved? There are many squamate genomes available (including at least one other species of gecko). This analysis should be relatively straightforward to perform and would not require the generation of additional data.

2) Sequencing a more diverse collection of leopard gecko samples might help distinguish which sequence alterations are unique to the Lemon Frost allele. Are there significant barriers to doing this?

3) In the last paragraph of the discussion, the authors’ state “We found that an evolutionarily conserved gene, SPINT1, regulates the proliferation of white iridophores in the leopard gecko.” While this statement is probably true, definitive evidence is not presented in the manuscript - neither protein coding changes nor expression changes in SPINT1 were identified. The authors should tone down this statement.

**Have all data underlying the figures and results presented in the manuscript been provided?**

Reviewer #1: None

Reviewer #2: Yes

PLOS authors have the option to publish the peer review history of their article (what does this mean?). If published, this will include your full peer review and any attached files.

Reviewer #1: No

Reviewer #2: No

---

## [Editor Report · Decision Letter 1]

4 May 2021

Dear Dr Kruglyak,

We are pleased to inform you that your manuscript entitled "Genetics of white color and iridophoroma in “Lemon Frost” leopard geckos" has been editorially accepted for publication in PLOS Genetics. Congratulations!

Yours sincerely,

Hua Tang

Section Editor: Natural Variation

PLOS Genetics

Scott Williams

Section Editor: Natural Variation

PLOS Genetics

Comments from the reviewers (if applicable):

**Data Deposition**

http://datadryad.org/submit?journalID=pgenetics&manu=PGENETICS-D-21-00137R1

**Press Queries**

---

## [Editor Report · Acceptance letter]

28 May 2021

PGENETICS-D-21-00137R1 

Genetics of white color and iridophoroma in “Lemon Frost” leopard geckos 

Dear Dr Kruglyak, 

We are pleased to inform you that your manuscript entitled "Genetics of white color and iridophoroma in “Lemon Frost” leopard geckos" has been formally accepted for publication in PLOS Genetics! Your manuscript is now with our production department and you will be notified of the publication date in due course.

With kind regards,

Katalin Szabo

PLOS Genetics

On behalf of:
